# Rotavirus A Genome Segments Show Distinct Segregation and Codon Usage Patterns

**DOI:** 10.3390/v13081460

**Published:** 2021-07-27

**Authors:** Irene Hoxie, John J. Dennehy

**Affiliations:** 1Biology Department, The Graduate Center, The City University of New York, New York, NY 10016, USA; john.dennehy@qc.cuny.edu; 2Biology Department, Queens College, The City University of New York, Flushing, New York, NY 11367, USA

**Keywords:** reassortment, phylogeny, virus evolution, genome constellations, genetic diversity, codon bias

## Abstract

Reassortment of the Rotavirus A (RVA) 11-segment dsRNA genome may generate new genome constellations that allow RVA to expand its host range or evade immune responses. Reassortment may also produce phylogenetic incongruities and weakly linked evolutionary histories across the 11 segments, obscuring reassortment-specific epistasis and changes in substitution rates. To determine the co-segregation patterns of RVA segments, we generated time-scaled phylogenetic trees for each of the 11 segments of 789 complete RVA genomes isolated from mammalian hosts and compared the segments’ geodesic distances. We found that segments 4 (VP4) and 9 (VP7) occupied significantly different tree spaces from each other and from the rest of the genome. By contrast, segments 10 and 11 (NSP4 and NSP5/6) occupied nearly indistinguishable tree spaces, suggesting strong co-segregation. Host-species barriers appeared to vary by segment, with segment 9 (VP7) presenting the weakest association with host species. Bayesian Skyride plots were generated for each segment to compare relative genetic diversity among segments over time. All segments showed a dramatic decrease in diversity around 2007 coinciding with the introduction of RVA vaccines. To assess selection pressures, codon adaptation indices and relative codon deoptimization indices were calculated with respect to different host genomes. Codon usage varied by segment with segment 11 (NSP5) exhibiting significantly higher adaptation to host genomes. Furthermore, RVA codon usage patterns appeared optimized for expression in humans and birds relative to the other hosts examined, suggesting that translational efficiency is not a barrier in RVA zoonosis.

## 1. Introduction

The high mutation rates and large population sizes of RNA viruses allow them to rapidly explore adaptive landscapes, expand host-ranges, and adapt to new environments. Segmented RNA viruses also may undergo ‘reassortment’ whereby viruses swap entire genome segments during coinfection [1]. Reassortment may allow rapid evolution of specific viral traits such as, for example, the acquisition of novel spike glycoproteins during the emergence of H1N1 influenza A in 2009 [2]. Similarly, reassortment among segmented dsRNA rotaviruses may have significant implications for human health [3], but it is challenging to determine the prevalence of rotavirus reassortment in nature. Our motivation here is to elucidate apparent restrictions (or lack thereof) to RVA genetic exchange in nature by comparing the relative linkage between each of the RVA segments as shown by phylogeny. In addition, we parse the evolutionary constraints that may contribute to the distinct phylogenies of each segment.

The rotavirus genome consists of 11 segments of double-stranded RNA, each possessing a single open reading frame, except for segment 11, which contains two genes. Rotaviruses are classified based on the antigenicity of the VP6 protein into groups A through I [4]. The consensus is that viruses from different groups cannot reassort with one another [5], though some rare cross-group reassortment events appear to have occurred [6]. While mammalian RVA strains routinely reassort with other mammalian RVA strains, reassortment between avian and mammalian RVA strains does not seem to occur outside of laboratories [7]. Nevertheless, there have been cases of mammalian strains transmitting between birds [8] (Figure 1), avian strains infecting mammals [9] (Figure 1) and causing encephalitis [10], evidence of an avian RVA isolate with a mammalian VP4 gene [11], and some in vitro reassortment assays [7,12], most recently confirming that mono-reassortments with avian segments 3 and 4 can be recovered using the SA11 reverse genetics system [13]. Reassortment between RVA strains from different mammalian species is also less common; however, it is not clear whether this lack of reassortment is due to biological incompatibilities as opposed to genetic incompatibilities. Genotypes resulting from interhost-species reassortment, while rare, have not only occurred but have fixed in populations [14,15,16,17,18,19,20,21,22], indicating that reassortment between even distantly related strains is potentially a significant driver of rotavirus evolution [23,24,25,26].

### 1.1. Rotavirus A Genome and Proteins

While the RVA genome is double-stranded, RVA RNA is packaged into and exits capsids as positive-sense, single-stranded RNA (+ssRNA). The RNA-dependent-RNA-polymerase, VP1, and the capping enzyme, VP3, are anchored to VP2 pentamers to form the replicase complex. This complex is present at each of the rotavirus capsid vertices through which +ssRNAs are released [27]. The inner VP2 layer is enclosed by an intermediate layer composed of VP6 protein, and the outermost layer is composed of the glycoprotein (VP7) and the multimeric spike protein (VP4) [28].

In addition to the six structural proteins, the RVA genome also encodes five to six non-structural proteins on five separate segments. The NTPase (NSP2) is a +ssRNA binding protein that forms a doughnut-shaped octamer with a positively charged periphery, allowing +ssRNA to wrap around and bind within its grooves during genome packaging [29,30]. NSP2 protein interactions with the +ssRNA are therefore critical to stabilizing +ssRNA contacts [31]. NSP2 also interacts with NSP5 to form viroplasms (along with other viral proteins) [32], where genome packaging, replication, assembly of progeny cores, and double-layered particles (DLPs) occurs [33,34,35]. To effect transmission, RVA encodes NSP4, an endoplasmic reticulum associated viroporin and enterotoxin, which elevates cytosolic Ca^2+^ in cells. This Ca^2+^ elevation ultimately results in diarrhea/vomiting in hosts [36,37,38,39,40]. NSP4 also interacts with VP6 on DLPs during RVA production [41,42,43,44]. The non-structural protein encoded on segment 7, NSP3, hijacks cellular translation and is a functional analog of the cellular poly (A)-binding protein [45,46,47]. NSP3 binds the group-specific consensus tetranucleotide *UGACC [45,46,48] located at the 3′ end of RVA ssRNA, which suggests inter-group reassortment does not occur. Lastly, NSP1 disrupts cellular antiviral responses [49], so may also play a role in the host-range [50]. Both segment 5 (NSP1), as well as segment 11 (NSP5/6) are especially prone to genome insertions [51,52,53].

For reassortment to occur, +ssRNAs from two or more parents must be packaged into the same virion. When reassortment occurs, the resulting reassortant virus must maintain the many protein and RNA interactions required for efficient packaging and replication. Even if the reassortment virus is functional, it may still be outcompeted by other genotypes, go extinct due to transmission bottlenecks, or evolve compensatory mutations. These factors, along with the high genetic diversity of RVA strains, make predicting reassortment facility or barriers to reassortment difficult.

### 1.2. Selective Pressures on Synonymous Sites

RVA is the most common cause of diarrheal disease in young children and is also an important agricultural pathogen, particularly for cows and pigs (Figure 1). RVA vaccines RotaTeq and Rotarix have produced substantial selective pressure on circulating RVA strains since they were first administered on a large scale in 2006 and 2008 respectively. This selective pressure seems to have favored certain genome constellations. Strains relevant to humans mostly consist of genogroup 1 (Wa-like) genes, genogroup 2 (DS-1-like) genes, or genogroup 3 (AU-1-like) genes. Specific G and P types are also associated more with specific genogroups [54,55].

Selective pressure on amino acid sequences to conserve protein interactions is a barrier to reassortment compatibility, but segmented viruses are also under considerable selective pressure on synonymous sites as RNA-protein and RNA-RNA interactions are critical in virus packaging [30,39,40,41,42]. RVA genome assortment requires +ssRNA molecules from each segment to form complexes with one another before being packaged, which relies on packaging signals on each segment.

Synonymous sites may also be under selective pressure for certain codon usage patterns. Codon usage bias results in varying levels of efficiency in translation, with highly expressed genes showing stronger bias for codons abundantly available in the host tRNA pool [56,57], and marginally expressed genes displaying less codon bias [58]. Codon usage adaptation to the host is a well-documented phenomenon in DNA viruses [57,59,60] and is observed in RNA viruses as well [61,62,63,64,65,66,67], though the constraints of a secondary RNA structure, as well as the high rate of mutation in RNA viruses, may lower the relative effects of translational selection in RNA viruses. Codon bias is also explained by drift and mutational pressure (i.e., bias towards A/T/U or G/C mutations) as well as translational selection [68]. Codon usage that is too similar to the host can also lower efficiency if viral proteins cannot fold properly [69], and differential codon adaptation can be a mechanism of controlling viral gene expression [70,71].

Rotaviruses have been shown to exhibit codon bias [72,73], but have especially divergent codon usage patterns from humans relative to other RNA viruses [74]. Codon usage can be a potential hindrance to zoonosis if a virus infects a host that cannot efficiently translate the virus’ proteins. RVA’s broad host range and ability to undergo genetic exchange makes RVA’s zoonotic potential a cause for some concern.

We show that RVA’s segments have distinct evolutionary histories, demonstrating the impact reassortment has had on mammalian RVA between the late 1950s to 2017. Because each RVA segment is under different selective pressure, we also tested whether there was evidence for translational selection on synonymous sites for each of the segments. To assess whether certain segments showed more codon adaptation against different common host genomes, indicating translational efficiency differences, we tested for neutrality in codon position 3, as well as variations in codon adaptive indices and relative codon deoptimization indices for each segment. To test whether RVA showed signs of codon optimization to a particular host, we compared strains isolated from specific hosts against their host genome and other RVA host genomes. We found differences in codon usage patterns between segments, with segment 11 (NSP5) having significantly higher codon adaptation to host genomes, however, our study indicates codon usage is not a barrier to rotavirus zoonosis.

## 2. Materials and Methods

### 2.1. Sequence Alignment and Phylogenetic Analysis

From all complete RVA genomes in NCBI’s Virus Variation Resource, 789 complete mammalian rotavirus A genomes isolated between 1974 and 2017 were chosen. To minimize sampling bias, we excluded any isolates where three or more genomes shared the same sequence identity for NSP4. To ensure that the analysis of each RVA segment employed the same set of strains, we filtered out the selected strains from files containing all available genomes for each segment in Python v3.9.2 (xml files available in Appendix A). The pooled sequences of each of the 11 RVA segments were independently aligned using MUSCLE v3.8.31 [75] and visually inspected for obvious sequencing errors or low-quality sequences. Sequences that were unusually long, short, or contained ‘N’ nucleotides were all removed. We performed phylogenetic analyses using BEAST v1.10.4 [76]. We used tip dating to calibrate molecular clocks and generate time-scaled phylogenies. We note that divergence-date estimating in viruses using tip-date calibration can be especially erroneous if there is a poor model fit to the data [77]. Genetic diversity, a contemporaneous bias of available virus sequences, lineage-specific variation in rates over time, long-term purifying selection, and inappropriate priors can result in substantial errors in the phylogeny and divergence date estimates [78,79,80]. We excluded Avian RVA from the phylogenetic analyses to minimize error in divergence-date estimates, which can be exacerbated with the inclusion of deeper nodes. As Mammalian RVA is endemic in the population, and has been for quite some time, this also somewhat minimizes the error associated with sampling bias.

The analyses were run under an uncorrelated relaxed clock model using a time-aware Gaussian Markov random field Bayesian Skyride tree prior [81]. Segment alignments were partitioned by coding sequence and untranslated region and run using a GTR + Γ + I substitution model and partitioned by codon position for the coding sequence partition. VP4 was partitioned by the VP5 * and VP8 * protein domains. Due to the large insertions in segment 11 (NSP5/6), this segment required three partitions based on insertion locations. Log files in Tracer v1.7.1 [82] were analyzed to confirm sufficient effective sample size (ESS) values. The alignments were run for three chains with a 500,000,000 Markov chain Monte Carlo (MCMC) chain length, analyzed on Tracer v1.7.1, and combined using LogCombiner v1.10.4 [83]. Trees were annotated with a 10% burn-in using TreeAnnotator v1.10.4 [82]. The best trees were visualized using FigTree v1.4.4, with the nodes labeled with posterior probabilities and node bars representing 95% confidence intervals for the divergence dates (annotated tree files available in Appendix A). Bayesian Skyride plots were made in Tracer v1.7.1 to compare each segment’s changes in effective population size (used as a proxy for relative diversity) since the root of the tree (~50 years prior to 2017 for most segments).

We used the R package ‘distory’ to calculate the geodesic distances between segments. Geodesic distance uses topology and branch length to visualize the tree space of the 11 segments to determine which segments share a close evolutionary history. To account for phylogenetic uncertainty, 350 post-burn-in, randomly chosen trees were sampled from the BEAST v1.10.4 tree file for each segment. We applied multi-dimensional scaling using the R package, ‘tree space’ to determine whether the ‘time to the most recent common ancestor’ (TMRCA) was consistent between segments. The correlation coefficient of TMRCA estimates from all pairwise comparisons of the 11 segment trees was used to estimate tree distance and then the matrix of tree distance was plotted. Variation in branch length between different segment trees was visualized as a cloud of points where the center represents the mean of several hundred trees. Segments that co-segregate overlapped in three-dimensional space, while segments that did not co-segregate were isolated in space.

To compare the segments’ relative host boundary conservation, we calculated the association index (AI) statistic [84] and parsimony score statistic (PS) [85] using the Bayesian tip-association significance testing program, BaTS [86]. To account for phylogenetic uncertainty 300 random post-burn-in trees for each segment were used for the analysis. As complete RVA genomes from non-human hosts are relatively under-sampled, for the 11 phylogenies used in the distance analysis, we assigned states for each isolate as being either from human or nonhuman hosts. We also created two separate phylogenies (Appendix A) using an additional, different set of isolates for segments 4 and 9, in which an effort was made to lower host-sampling bias and maximize the RVA genetic diversity captured. The phylogenies consisted of 127 non-human isolates which had complete sequences available for at least segments 4 and 9. We calculated the phylogenetic trait association for each of the two segments’ phylogenies with regards to specific host species, to test if the same pattern of relative host-species association was still observed between the two segments when using a less-biased host-sampling.

### 2.2. Codon Bias Analysis Comparison

The relative codon deoptimization index (RCDI) was used to assess if the codon usage of a gene was similar to the codon usage of a reference genome (2953 coding sequences for *Sus scrofa* (pig), 93,487 coding sequences for *Homo sapiens* (human), 6017 coding sequences for *Gallus gallus* (chicken), 13,374 coding sequences from *Bos taurus* (cow), 1194 coding sequences for *Canis familiaris* (dog), and 1115 coding sequences for *Oryctolagus cuniculus* (rabbit)) [87]. RCDI values range from 0.0 to 1.0 with 1.0 indicating maximum codon usage compatibility with a reference genome. Similarly, the codon adaptation index (CAI) was used as a measure of codon usage adaptation to the most used synonymous codons of a reference genome and was used to predict the expression levels of genes [87]. CAI values range between 0.0 and 1.0, where higher CAI values for a particular reference set indicate higher expression levels. To determine if there were significant differences in CAI/RCDI values between the different segments, we calculated CAI and RCDI values using the CAIcal server (http://genomes.urv.es/CAIcal/ (accessed on 15 August 2020)) for each segment using a subset of RVA isolates containing multiple representatives of common mammalian RVA genotypes [88]. Statistical analyses were performed to assess whether RVA was genetically compatible in its codon usage patterns with a set of RVA host reference genomes. To reduce bias that would result from analyzing a small number of genes, the RVA host reference genomes were chosen based on the availability of a large number of genes analyzed for their codon usage tables in the Codon Usage Database (http://www.kazusa.or.jp/codon/ (accessed on 15 August 2020)). Reference sets for chicken, human, pig, dog, rabbit, and cow genomes were used for the analysis. We performed Tukey’s Honest Significant Difference (HSD) test to compare mean CAI and RCDI values between segments. We also used Tukey’s HSD tests to compare mean CAI and RCDI values between different host genomes after combining all segments’ values.

To test for neutrality at the third codon position, a neutrality plot was made by comparing the GC contents at the first, second, and third codon positions. GC12 being the average of GC1 and GC2 was plotted against GC3. If GC12 and GC3 are significantly correlated to one another, and the slope of the regression line is close to 1, mutational bias is assumed to be primarily responsible for shaping codon usage patterns rather than translational selection. Selection against mutation bias can lead to larger differences in GC content between positions 1 and 2, and position 3 and little to no correlation between GC12 and GC3 [89].

RVA strains isolated from specific hosts (pig, human, cow, and avian) were also compared to the pig, human, cow, and chicken genomes to test whether RVA genomes showed evidence of adapting to specific hosts. Seven complete genomes (77 segment genomes) for each host, isolated from either avian, cow, human, or pig hosts. While reassortment and spillover events occur in RVA evolution, there are larger generally bovine/porcine/human/avian clades (Figure 1, Appendix A), so isolates representing their general host strain clades were chosen. If RVA genomes did not appear to differ in CAI and RCDI patterns based on host type, then it would suggest mutational selection was the dominating force over the translational selection, and that selection at the translation level was weak or undetectable.

## 3. Results

### 3.1. Different Tree Space Occupied by the 11 Segments

Multi-dimensional scaling of the random, post-burn-in sampling of BEAST trees for each of the 11 segments revealed that segments 4 and 9 occupy distinct tree spaces from each other and the rest of the genome (Figure 2). Segments 10 and 11 occupied indistinguishable tree space indicating close geodesic distances and high levels of evolutionary linkage between them. Segments 3 (VP3), 5 (NSP1), and 6 (VP6) also shared highly overlapping tree space with one another. Segment 2’s (VP2) evolutionary history was most like segment 6 (VP6). None of the segments overlapped with segment 1 (VP1) except for segment 7 (NSP3). Segment 7 also had the most phylogenetic uncertainty of all 11 segments as shown by the larger spread around the plot of the post-burn-in trees in Figure 1. The best-supported trees are depicted for segments 1–3 (Figure 3), segments 4–6, 9 (Figure 4), and segments 7, 8, 10, and 11 (Figure 5), with the host species coded on the tree and the branch lengths color-coded by relative evolutionary rate (Figure 3). While segment 4 had a more independent evolutionary history from other segments, its tree and AI/PS statistics suggested this segment has stricter host boundaries than segment 9 (Figure 4, Table 1), indicating either less opportunity for divergence due to selection responses to host species change or that segment 4 has a stronger role in host determination.

The AI and PS statistics calculated from the post-burn-in random trees (Table 1, Figure 2, Figure 3, Figure 4 and Figure 5) comparing human to non-human host spillover events, indicated segment 4 was least likely to switch hosts (AI = 1.23, PS = 11.71) while segment 9 was the most prone to switching hosts (AI = 3.83, PS = 25.80). The AI and PS statistic for separately tested non-human host phylogenies (Appendix A) using each specific host species as a state, was 3.20 for segment 4 and 4.18 for segment 9. A lower AI statistic indicates a stronger correlation between trait and phylogeny, so this result also indicates VP7 has less strict host boundaries relative to the other segments. We note that as the 11 trees used for the whole-genome comparison were under-sampled for non-human hosts, a larger analysis should be done with less-biased host sampling before confident conclusions can be formed regarding a comparison of relative host associations between all of the 11 segments.

### 3.2. Evolutionary Rates

Segment 8 (NSP2) displayed the lowest mean substitution rate (1.48 × 10^−3^ substitutions per site per year), while segment 4 (VP4) had the highest (3.77 × 10^−3^ substitutions per site per year) (Table 2). The mean rate for segment 11 was likely skewed higher due to the frequency of large insertions into the segment. Segments 1–3 showed similar evolutionary rate changes at corresponding clades and time periods in their trees, with the higher evolutionary rates occurring earlier in their evolutionary histories (Figure 3) (for node confidence intervals at divergence dates see Appendix A). Higher evolutionary rates tended to be observed along branches leading towards non-human host isolates, particularly for VP7, which had more rate variation across the tree than the other segments (Figure 4). For example, the *Ailuropoda melanoleuca* (giant panda) strain (represented by the green asterisk in Figure 3, Figure 4 and Figure 5) possesses a genomic backbone that is within a cluster of pig and cow strains, except for segment 9. The giant panda’s RVA segment 9 occupies a more divergent branch associated with a higher evolutionary rate than the rest of its segments.

Coefficients of variation (CoV) for each segment (Table 2) were consistently high, supporting the assumption that a strict molecular clock is inappropriate for this analysis, and that a relaxed clock is a better choice. All segments exhibited relatively similar TMRCA estimates with segments 1–3 possessing slightly older TMRCA dates than the other segments (Table 2). Segment 1 had the oldest TMRCA estimate (~1957) of all the segments, while segment 11 had the most recent TMRCA estimate (~1969). Node bars for 95% confidence intervals of the node divergence dates are shown in Appendix A for each segment. Since this data set of 789 genomes includes strains collected from geographic areas across the world, it appears that most of the mammalian RVA genetic diversity present today has evolved in the past ~60 years.

The Bayesian Skyride plots indicate that RVA segments reached their peak diversity levels around the year 2000, with a steep decline around the year 2007, which coincides with the introduction and broad-scale use of RVA vaccination (Figure 6).

### 3.3. Differing Codon Usage Patterns by Segment

Compared to the other ORFs, NSP5 possessed significantly higher CAI scores and significantly lower RCDI scores across all host genomes. The wide range of values for each RVA segment suggests that, while there may be differences in selective pressure on codon usage by segment, the translational selection was relatively weak compared to mutational selection for the rest of the genome.

Codon usage patterns for both mammalian and avian RVA appear more compatible with avian genomes than mammalian genomes (Figure 7 and Figure 8), however, human genomes and avian genomes showed similar compatibility between one another, with RVA genomes, and their CAI scores were not significantly different. RVA had higher CAI values and RCDI scores closer to 1 for both human and avian genomes relative to rabbit, cow, pig, and dog genomes. RVA compared against the pig genome resulted in the lowest CAI scores for all segments and strains. In other words, based on CAI and RCDI metrics, RVA is predicted to be the least translationally efficient within pigs.

A neutrality plot revealed that NSP4 had the highest GC content in position 3 (GC3) relative to the other segments. VP6 had the highest GC content in positions 1 and 2 (GC12) relative to the other segments while NSP1 and VP3 had the lowest GC12 content. While VP6’s GC12 content was higher than the rest of the RVA genome, VP6’s GC3 content was not. The slopes for all regression lines deviated from 1 (Figure 9), ranging from 0.043 (VP7) to 0.316 (NSP1), indicating that there was significant selective pressure on position 3 for codon usage for all segments. The slopes indicate NSP1 is under more mutational pressure than the other segments, while VP7 is under more selective pressure at position 3 compared with the other segments, however, the lower GC3 values relative to GC12 in VP7 actually indicates *lower* adaptation to the host genome.

The lower GC3 values relative to GC12 in VP1 and VP3 also showed low slopes (0.06 and 0.05 respectively), suggesting that they may also be under less mutation pressure and more translational selection, however like VP7, the relatively low GC3 content in already high AU genome, suggests the translational efficiency would be lower. The correlation coefficient overall for GC12 and GC3 was 0.261 (*p* < 0.001). While some segments/ORFs (VP1, VP2, VP4, NSP1, NSP2, NSP3, and NSP5) were more significantly positively correlated between GC12 and GC3, segments 9 and 10 displayed no significant correlation.

### 3.4. Synonymous Sites under Selection, However Evidence Does Not Suggest Translational Selection to Specific Hosts

When comparing strains isolated from different species, there was no evidence to support the conclusion that RVAs adapt their codon usage to specific hosts (Figure 10). While the bovine strains and avian strains had RCDI values closer to 1.0 than the pig and human RVA strains, there were no significant differences between RCDI values of avian isolates compared to avian genomes vs. RCDI values of bovine isolates compared to avian genomes. Based on RCDI values, bovine strains were “most compatible” with avian genomes, and human strains were “less compatible” to the human genome than to bovine strains. Given that avian and mammalian RVAs do not exchange genetic information in nature, there is a substantial divergence between avian RVA isolates and mammalian RVA isolates. The fact that RVA strains show more similar codon patterns between bovine and avian strains than between human and bovine strains suggests that translational selection by the host does not play a large role in codon bias.

Rotavirus genes overall had higher CAI scores and lower RDCI scores when contrasted with the avian genome codon usage patterns (Figure 7). The lowest CAI and highest RDCI scores were observed when RVA genes were compared with rabbit genomes. However, there was little variation depending on which host the viral strain was isolated from (i.e., avian RVA strains did not have significantly different scores when compared to bovine RVA strains using the same reference codon usage patterns) (Figure 10). There was no evidence for detectable translational selection by the host, however, the observation that RVA was generally less optimized for the non-human mammalian genomes suggests that RVA may have higher protein expression levels in humans.

## 4. Discussion

While segment exchange between different RVA genotypes is common, reassortment is not a random process [90,91,92,93,94]. However, the limits of segment exchange, whether ecological or mechanical, are poorly understood. Some segment combinations work well together, whereas others are incompatible [95]. Numerous factors potentially could affect whether segments from different RVA genotypes are able to reassort, including protein interactions, RNA-RNA and RNA-protein interactions, and the need to maintain host range and ensure RNA packaging efficiency.

The goal of this study was to better predict potential (or unlikely) genome constellations that may emerge in nature and enhance our understanding of why some segments co-segregate, and some do not. To this end, we compared the evolutionary histories as well as some of the selective pressures acting on the 11 RVA segments, first by performing phylogenetic analyses on a large collection of complete RVA genomes, and then by assessing the selective pressures acting on synonymous sites by segment and host type. We estimated the diversity levels of each RVA segment over the last ~60 years and linked decreases in diversity to the introduction of RVA vaccines between 2006 and 2008.

The segments that share highly similar tree space may share important interactions that rely on a higher percentage of the sequence (e.g., selective pressure at synonymous sites), while the segments that inhabit distinct tree space are likely more flexible on a variety of genetic backgrounds. RVA’s RNA segments are under different evolutionary pressures, which is clear from the distinct evolutionary histories of the segments (Figure 2), and the significant difference in nucleotide composition between the segments, despite coming from the same genome (Figure 7 and Figure 9). The zoonotic potential for rotaviruses makes understanding restrictions on genetic exchange important, as outcrossing events can result in novel strains which may cause more severe disease or be better able to evade a vaccine or immune response. Both codon usage and reassortment potential can be important factors in the viral host range, and both are constrained by RNA interactions, translational selection, and mutational bias. Understanding the relative constraints of the RVA genome can help better assess the risk of zoonotic outbreaks and emerging strains.

### 4.1. Potential Impact of Rotavirus RNA and Protein Interactions on Segment Co-Segregation 

RVA protein and RNA molecules interact with each other in a variety of ways during infection and assembly processes. Incompatibilities among different genotypes resulting from these interactions may limit the diversity of genome constellations observed in nature. For example, RNA secondary structures and segment-specific sequences found in the non-translated terminal regions (NTRs) may govern the formation of the supramolecular RNA complex associated with segment packaging [31,94,96,97,98,99] Sequence mismatches between segments from coinfecting RVAs may prevent segment interactions and co-packaging, and hence, the generation of reassortment RVAs. One of the studies [96] also suggested that VP4 has a less conserved terminal RNA structure, so the importance of these RNA interactions may vary significantly by segment.

The order in which the segments associate to form the supramolecular RNA complex may be sequential. In this scenario, the smallest segments interact first, then they recruit intermediate-sized segments before finally incorporating the largest segments [97]. In addition, incompatibilities in the 3′-NTRs of the smaller segments may have stronger effects on segment co-segregation than do the larger segments. For instance, the smallest RVA segments, segments 10 and 11 must directly interact before they can interact with larger segments. Evidence for this supposition that the smaller segments’ RNA structure is under strong selective pressure along with its protein structure (i.e., synonymous mutations are often not neutral) also comes from our observation that segments 10 and 11 co-segregated with one another more strongly than other pairs of segments (Figure 2), despite their many protein interactions with other segments.

There is evidence for frequent inter-host-species reassortment of segment 10, which encodes the enterotoxin, NSP4, in nature [15,100], so reassortment of segment 10 may be more dependent on the sequence conservation of terminal +ssRNA between segments 10 and 11, than on protein interactions. In addition to NSP5, segment 11 also encodes a second out of frame protein, NSP6, via leaky scanning. NSP6 is not required for virion function and is sometimes not expressed in rotaviruses [101]; however, it is constrained due to overlapping with NSP5. Segment 11’s RNA structure is thought to have some functional importance [96,102] however segment 11 appears more tolerant of genome insertions than other segments [51,52], likely due to packaging signal duplication [52]. The close evolutionary histories of segments 10 and 11 may also relate to NSP4-NSP5 interactions during viroplasm formation [34].

The tree spaces of both segments 4 and 9 (Figure 2) were notably distinct from the tree spaces of the rest of the genome. RVA segment 4 encodes the spike protein, VP4, which is cleaved by trypsin into VP8 * and VP5 *. VP4 interacts with different receptors depending on the strain, including sialoglycans and histo-blood glycans [103,104,105] These different receptors partly explain why certain P types tend to dominate in different populations, species, and age groups [105,106]. Our results showed that the gene tree of segment 4 was distinct from the rest of the genome, suggesting that segment 4 may reassort more readily than other segments. Alternatively, segment 4’s divergent history could also be explained by VP4’s role in host determination (Table 1). Segment 9 (VP7) had the weakest association with host species (Table 1) which may explain its distance from VP4 and the rest of the genome. Other environmental studies have also found that segment 4 and segment 9 are more likely to appear in different genetic backgrounds [107,108,109]. Based on the high genetic diversity of segments 4 and 9, and segment 4 seeming to have a less conserved role in RVA +ssRNA assortment, reassortment into new genetic backgrounds may confer a selective advantage and a broader host range, as strains can have an opportunity to evolve in a novel host, they may otherwise be unable to infect, due to a barrier caused by another segment (e.g., poor receptor-binding). Segment 4’s larger geodesic distance from the rest of the genome and high diversity in RNA secondary structure, suggests that segment 4’s synonymous sites may be less critical to the segment’s function. Segment 9, on the other hand, is critical for the formation/stabilization of the supramolecular RNA complex and for packaging the genome. Both segments 4 and 9 have been shown to tolerate homologous recombination among highly divergent genotypes, including recombination events resulting in the disruption of many amino acids, whilst still maintaining overall tertiary structure [110].

Due to the importance of VP1, VP2, and VP3 during the formation of the virion, synthesis of dsRNA, and associating with the 11 +ssRNA segments, one might expect these segments to be the least likely to reassort independently. However, the critical VP1, VP2, and VP3 interactions are mostly protein-protein interactions, so even a genetically distant strain could maintain a conserved amino acid sequence allowing these segments some flexibility with their genetic background. Our results showed that, while these three segments are generally associated with the larger “gene tree” of the rest of the genome, they have a more independent evolutionary history than for example, segments 10 and 11, which almost entirely overlap in tree space.

Interestingly, segment 7 (encoding NSP3) shared the closest evolutionary history with segment 1 (Figure 1). We expected segment 1 to have the closest evolutionary history with segments 2 or 3 given their proteins’ interactions, however, VP1’s high degree of structural conservation [5] and having less functionally important RNA structure than the other segments may explain its tolerance for novel genetic backbones. Segment 7/NSP3 may have less strain-specific interactions with other segments resulting in less fitness variation following a reassortment event. Segment 7 may have endured a significant reassortment event around 1970 (Figure 5, and Appendix A) which also may explain its geodesic from the other segments. NSP3’s primary function is to recognize a conserved group-specific sequence present on all group A RV segments and interact with host eIF4G, a protein that is highly conserved among orthologs [111]. This suggests NSP3 genes could be flexible to many RVA genetic backgrounds and hosts.

Segment 5 (NSP1) was only particularly distinct in the tree space (Figure 2) from segments 1, 4, and 9. This was somewhat surprising as NSP1 has relatively low conservation, can tolerate insertions and deletions, and is not required for rotavirus replication in vitro (although the RNA is still required for packaging). NSP1’s important role in targeting the host’s antiviral response as an interferon antagonist [112], inhibiting apoptosis [113,114], and activating NFkappaB [115] may explain a stronger host-association. NSP1 reassortment with different host strains may confer a deleterious effect in vivo, despite the reassortment’s ability to compete in vitro (i.e., in the absence of a significant immune response).

### 4.2. Rotavirus Evolution Following the Introduction of Rotavirus Vaccines

The live-attenuated pentavalent vaccine, RotaTeq (Merck, West Point, U.S.), was introduced in 2006, and the live-attenuated monovalent vaccine, Rotarix (GlaxoSmithKline, Rixensart, Belgium, U.S.), was disseminated in 2008. RotaTeq contains five human-bovine reassortment viruses with strain serotypes of G1P [5], G2P [5], G3P [5], G4P [5], and G6P [8]), while Rotarix is comprised of the human G1P [8] strain, RIX4414. These vaccines provided effective protection against contemporaneous globally dominant strains G1P [8], G2P [4], G3P [8], G4P [8], and G9P [8]. Our results show a coincident decline in the relative diversities and effective population sizes of all RVA segments after 2006. Supporting this analysis, several studies, including an analysis of G12 strains in Africa, G12 strains in Spain, and a lineage of P [8] strains also show a general decline in diversity/effective population size after 2008 [116,117,118].

The introduction of the RVA vaccines resulted in a global reduction in RVA-associated mortality. However, vaccine effectiveness varied substantially by region [119] and resulted in changes to the circulating strain prevalence. For example, in a post-vaccine era, the prevalence of G9P [8], G2P [4], and G9P [4], G9P [8] increased, while the prevalence of G1P [8] and G3P [8] declined [120,121,122,123]. The emergence of rare genotypes or animal RVA reassortments in children also appears to be connected to the selective pressure imposed by vaccination. For example, the increase in abundance of G12 and G11 genotypes [121,122] and the appearance of atypical Wa-like and DS-1-like reassortments, such as the emergence of G1P [8] with a DS-1-like backbone in Malawi [124], appear linked to increases in RVA vaccination. Given DS-1-like and Wa-like segments are thought to have incompatibilities with one another, limiting their reassortment potential [95], the vaccine-induced selection for mutations can be difficult to assess. That is, it is difficult to determine whether emerging fixed mutations are the direct result of escape mutants or are compensatory mutations resulting from novel reassortments. A study on G2P [8] evolution did not observe evidence of vaccine-induced selection; however, another study focusing on P [6], P [4], and P [8] genes did report substantial divergence from the vaccine strains.

Vaccine-induced selective pressure may partly explain the pattern observed when comparing the geodesic distances of the segments. Like the present study, which found that segments 4 and 9 (encoding serotype proteins VP4 and VP7) were especially amenable to reassorting into new genetic backgrounds, a study of European Bluetongue virus (BTV) isolates also found that segments 2 and 6 (encoding the BTV homologs of VP4 and VP7) were quite distant from the rest of the genome in tree space [125]. However, the BTV segments 2 and 6 were closer to one another in tree space, whereas in RVA segments 4 and 9 were highly distinct from each other. Additionally, BTV segments 7 (encoding the inner capsid protein) and 10 (encoding NS3) shared close evolutionary histories with one another, and distinct from the rest of the genome. Interestingly, in another BTV study on strains from India, segment 4 (encoding a protein homologous to VP3 in RVA) was found to be the most isolated segment in tree space [126]. While there are BTV vaccines available, BTV does not infect humans, is not as globally distributed as RVA, and far fewer serotypes circulate, so there may be more selective pressure on RVA serotype segments to reassort.

While RVA segments were unlinked and had different phylogenies and rate variations along with their trees, the segments’ patterns in relative diversity over time mostly matched each other. In the Rift Valley fever virus [127], which is a three-segmented -ssRNA arbovirus, different skyline plots were observed for each segment, suggesting that the segments are evolving independently to some extent. In the Rift Valley fever study, the medium segment had a much larger effective population size than the small and large segments, indicating that the medium segment experienced reassortment events more frequently than the small and large segments, which tended to co-segregate. The differences in evolutionary patterns between RVA and Rift Valley fever virus may be a consequence of their differing epidemiology. Frequent outbreaks of Rift Valley fever are limited to sub-Saharan Africa, and their transmission relies heavily on mosquitos and not human-to-human transmission. Rotavirus is conversely, a globally present pathogen with many dominating strains constantly circulating amongst humans, likely making it less sensitive to bottleneck effects.

A more thorough comparison of the geodesic distances between the spike and outer capsid proteins and the rest of the genome in other dsRNA viruses is indicated. It would be interesting to ascertain whether the patterns observed in RVA are also seen in other dsRNA viruses. While both G and P types are significantly associated with certain hosts (Figure 1, Table 1), this study suggests segment 9 is the most flexible to different hosts, while segment 4 is the least. Rotavirus disease is typically discussed in terms of ‘GXP[Y]’ strains, however, segment 4 and segment 9 evolve independently both from each other and from the rest of the genome. The prevalence of common G and P combinations in association with certain backbones seems to have more to do with an ecological abundance of those strains and less to do with a functional constraint on the virus. As this study analyzed strains from prior to 2017, the global declines in RVA diversity in response to RVA vaccination may change.

### 4.3. Codon Bias Analysis Shows Codon Usage Differs between Segments, but Not between Host Strains

We contrasted the codon usage patterns of several common mammalian RVA genotypes with a set of RVA host reference genomes. Our results showed that mammalian and avian RVA codon usage patterns were most compatible with avian genomes (Figure 7). For example, the highest CAI values and lowest RCDI values in our comparative analysis suggested that RVA NSP5’s codon usage pattern was best adapted to the chicken genome (Figure 7). This finding suggests that RVA originated as an avian virus that subsequently expanded its host range to include mammalian hosts. The fact that RVA appears better adapted to human and avian genomes rather than the other mammalian genomes tested further suggests that RVA spread to other mammalian hosts after adapting to humans, though more evidence is required to confirm this hypothesis (Figure 8). Additionally, NSP5’s relatively high CAI value and low RCDI value (Figure 7; both values approach 1) may indicate a strong selective advantage for NSP5’s codon optimization. Since efficient NSP5 protein expression is critical for RVA viroplasm formation and replication [128,129,130], the higher CAI value may indicate that NSP5’s codons are optimized to match avian hosts to maximize NSP5 expression in host cells. The variation in GC content between segments was also notable, as being from the same genome, this suggests mutational bias towards AU or GC is not entirely responsible for RVA’s observed codon bias. The low GC3 content and especially low GC3:GC12 ratio seen in VP3 (Figure 9), in contrast with the high GC3 content of NSP5 and NSP4 (relative to the rest of the genome), could suggest it is beneficial for the rotavirus to have lower-efficiency VP3 expression relative to NSP4 and NSP5, during infection. Varying the codon bias by gene, or maintaining suboptimal codon bias to the host, may sometimes be beneficial [69,74], a phenomenon which has been observed for example, in hepatitis A virus [61].

Although the bovine and avian RVA strains sampled tended to have higher CAI scores and RCDI scores closer to 1, there was no evidence of divergence in codon usage among bovine and avian strains. The similarity in codon usage between host strains is in contrast with a study of influenza A which found that avian and human influenza strains have distinctly different G/C vs. A/U contents from one another [131]. Our finding suggests that, while RVA strains may indeed experience an advantage from matching the codon usage patterns of their hosts, it is unlikely that translational selection can counteract nucleic acid selection (i.e., selection favoring synonymous substitutions improving virus survival and reproduction). The wide range of CAI/RCDI values and GC3 content for some of the segments suggest that translational selection is not an especially strong selective pressure for every gene. However, translational selection does seem to be a stronger selective pressure at least in NSP5, based on its significant divergence from the rest of the RVA genome’s codon usage (Figure 9). The neutrality plot also shows that GC percentages at position 3 frequently are significantly different from the GC percentages at positions 1 and 2, which would indicate that strong selection is occurring at position 3 (Figure 9). The variation in GC content and codon usage by segment could point to RVA using codon bias as a mechanism of controlling viral gene expression. While there is no significant difference in codon usage between strains isolated from different hosts (Figure 10), notably RVA, in general, is significantly more optimized to human and avian genomes, than pig and cow genomes. This indicates translational efficiency is not a barrier for zoonosis between avian and mammalian strains.

There are some limitations in the assessment of codon usage patterns. Despite the potential advantage of possessing a codon usage pattern that strongly resembles that of a host, codon usage patterns on their own are inadequate for making inferences or conclusions about the selective forces acting on virus populations. Furthermore, a significant amount of codon usage pattern variation exists between genes of the same host, so forming conclusions regarding viral codon-level adaptation to the host is especially difficult. That is, having high CAI values or RCDIs close to 1.0 does not necessarily mean that the virus is more adapted to that host. It could, however, provide evidence that the viral genes are better expressed in a particular host or that a specific gene of a virus may be more efficiently expressed. Additionally, we note that rotaviruses have especially AU-rich genomes, and their codon usage patterns diverge from human usage patterns more than other human viruses [74]. It would appear that the benefits of being AU-rich outweigh any benefits conferred through codon optimization.

This study further supports caution when measuring for selection by comparing dN/dS ratios for rotaviruses, as the selective pressure at synonymous sites varies significantly by segment. While the evidence did not support RVA nucleotide composition or translation selection varying based on the host strain, certain segments were under stronger selectional (translational) rather than mutational pressure at codon position 3. RVA’s indistinctive codon usage by host strain, suggests translational efficiency is not an important host-range barrier for RVA.

## Figures and Tables

**Figure 1 viruses-13-01460-f001:**
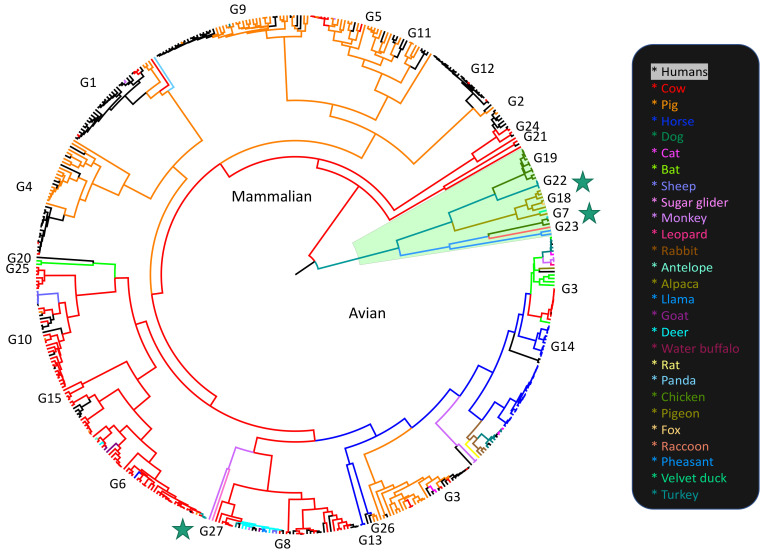
Phylogeny of segment 9 (VP7) of Rotavirus A colored by host species. Stars indicate avian-mammalian RVA spillover events (avian RVA to fox, avian RVA to raccoon, bovine RVA to turkeys). The alignment for this tree was made using complete VP7 sequences representing all known, available G types and hosts, to display the known-host range and host-boundary patterns for RVA. The VP7 (G) genotypes are labeled by clade. The tree file is available (Appendix A) with accession numbers for all sequences used. The phylogeny was run in BEAST v1.10.4 using a birth-death tree prior under an uncorrelated relaxed clock and a GTR+I+G substitution model, and the alignment was partitioned by translated and non-translated regions. Branch lengths correspond to substitutions.

**Figure 2 viruses-13-01460-f002:**
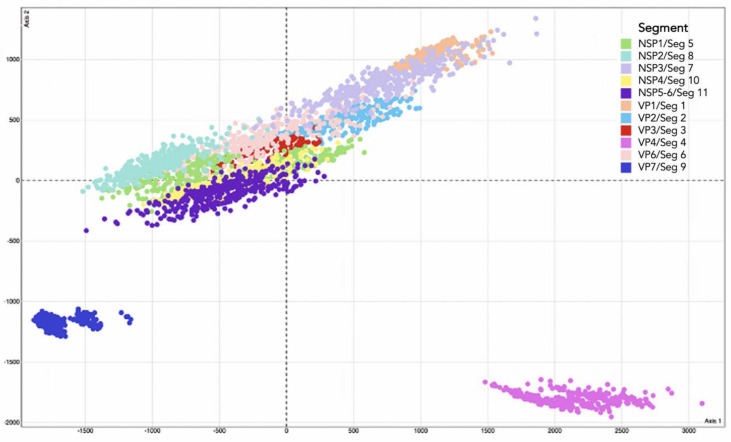
Multi-dimensional scaling plot using 350 post-burn-in BEAST trees from each of the 11 RVA segments. Randomly sampled post-burn-in trees were taken from each segment to account for phylogenetic uncertainty. Points sharing the same color are from the same segment as shown in the legend. Points closer to each other indicate close geodesic distances and high levels of evolutionary linkage.

**Figure 3 viruses-13-01460-f003:**
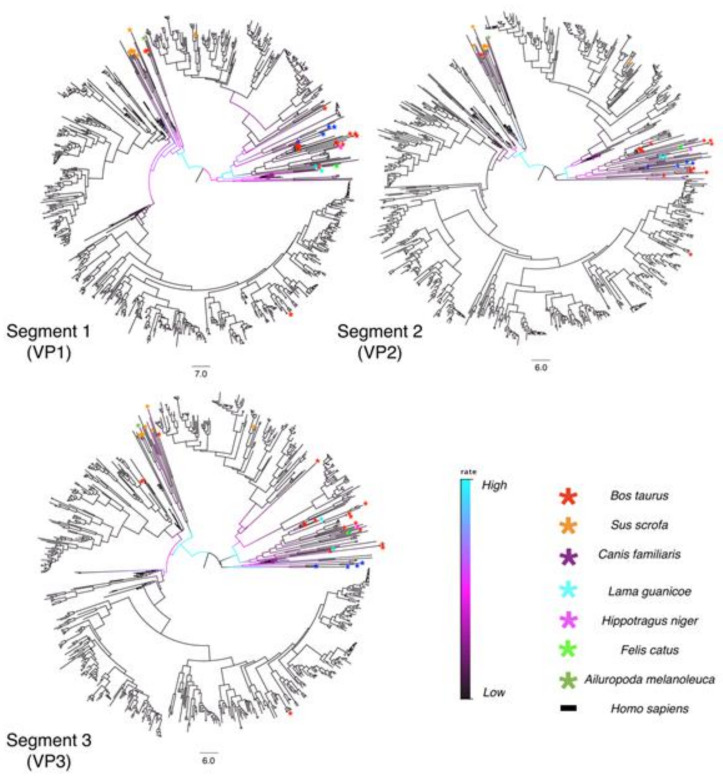
Time-scaled phylogenetic trees of Segments 1–3 for 789 mammalian RVA strains. The phylogenies are time-scaled using tip-dating. Scale bars below each tree represent branch length time in years. The branches are colored by rate. Cyan indicates the fastest evolutionary rate among all lineages, and black represents the slowest rate of evolution. Colored asterisks specify the host species the strain was isolated from as shown in the legend. Posterior probabilities, node bars for confidence intervals of the divergence dates, and tip labels for the strain names can be viewed by opening Appendix A.

**Figure 4 viruses-13-01460-f004:**
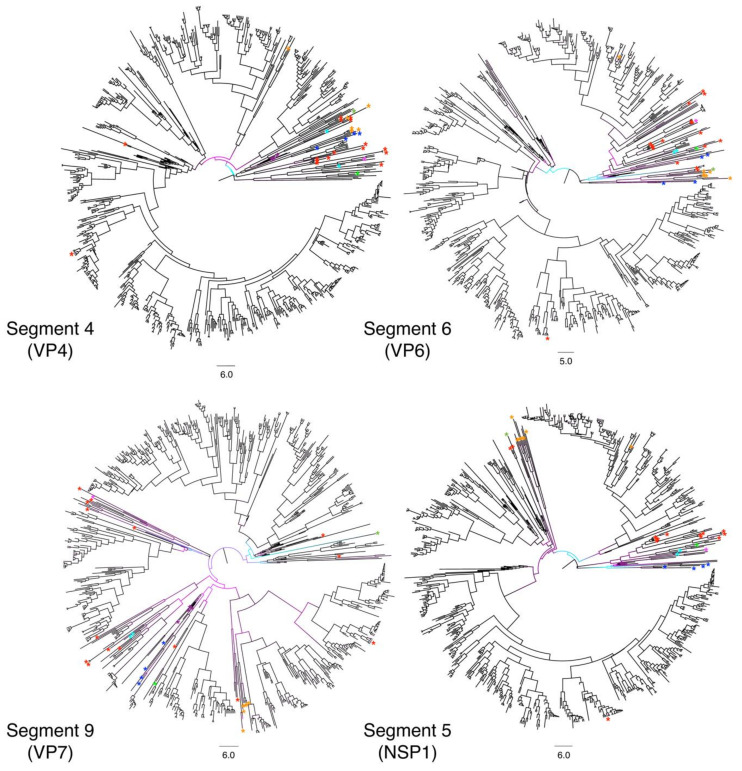
Time-scaled phylogenetic trees of Segments 4, 6, 9, and 5 for 789 mammalian RVA strains. Phylogenies are time-scaled using tip-dating. Scale bars below each tree represent branch length time in years. The branches are colored by rate. Cyan indicates the fastest evolutionary rate among all lineages, and black represents the slowest rate of evolution. Colored asterisks specify the host species the strain was isolated from as shown in Figure 3. Posterior probabilities, node bars for confidence intervals of the divergence dates and tip labels for the strain names can be viewed by opening Appendix A.

**Figure 5 viruses-13-01460-f005:**
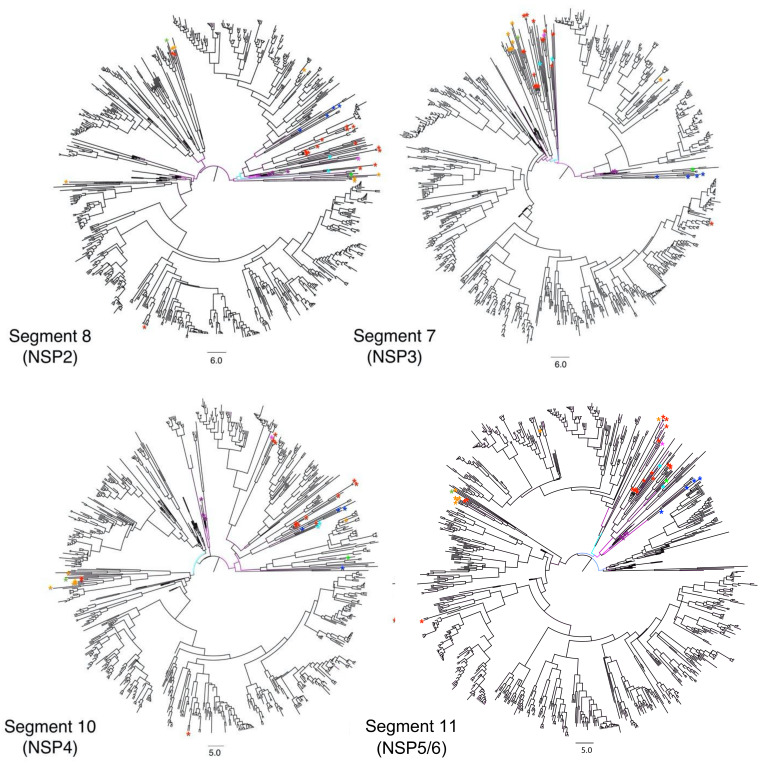
Time-scaled phylogenetic trees of Segments 8, 7, 10, and 11 for 789 mammalian RVA strains. Phylogenies are time-scaled using tip-dating. Scale bars below each tree represent branch length time in years. Table 1. Posterior probabilities, node bars for confidence intervals of the divergence dates and tip labels for the strain names can be viewed in Appendix A by opening Appendix A.

**Figure 6 viruses-13-01460-f006:**
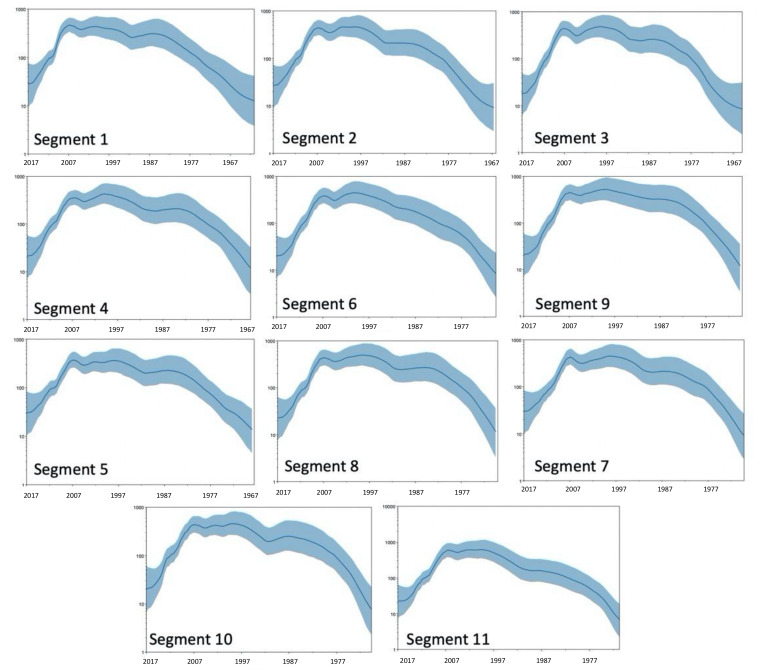
Skyride plots of the 11 RVA segments. The X-axis represents the date. The Y-axis represent effective population size and is a proxy for genetic diversity. All segments show a sharp decrease in diversity roughly 10 years before 2017 (2006–2008).

**Figure 7 viruses-13-01460-f007:**
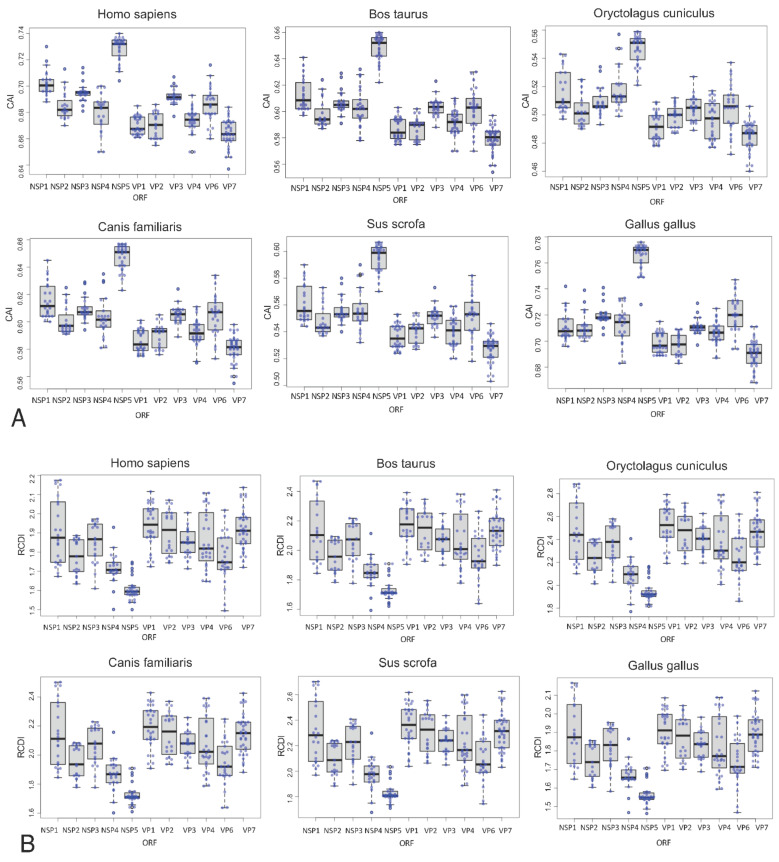
Boxplots of the Codon Adaptation Index and Relative Codon Deoptimization Index values for RVA genes with respect to different RVA hosts. (**A**). Sampled sequences representing common genotypes for each of the 11 segments were measured for CAI with respect to the six hosts shown. The Y-axis value ranges are different for each host as some hosts have higher CAI values. Higher CAI values indicate more efficient expression. (**B**). RCDI values for the same sampled sequences from A plotted for each segment with respect to each host. RCDI values closest to 1 indicate more similar codon usage patterns. The Y-axis value ranges are different for each host. Tukey-Kramer 95% pair-wise confidence intervals are shown in Appendix A.

**Figure 8 viruses-13-01460-f008:**
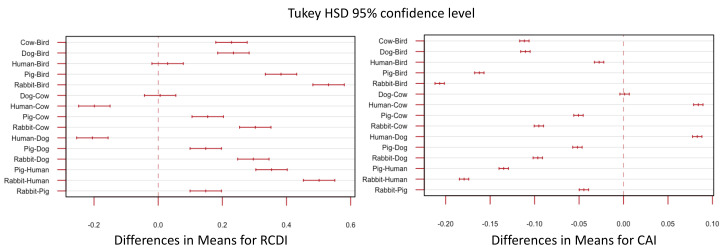
Tukey’s honest significance test for RCDI and CAI values between different host genomes. Values used for this test are the combined values for all segments used in Figure 6.

**Figure 9 viruses-13-01460-f009:**
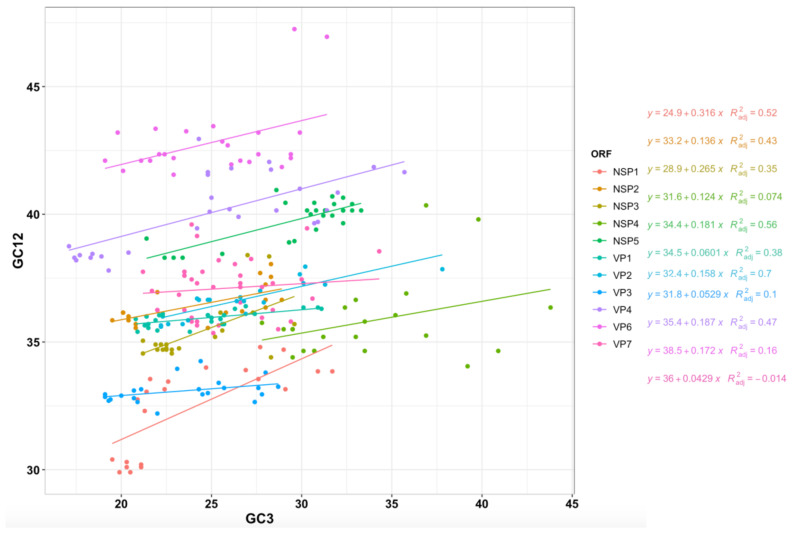
Neutrality plot of RVA ORFs. GC percentages of codon position 3 plotted against GC percentages of codon positions 1 and 2 (GC12) for the ORF for 11 RVA genes. Slopes significantly deviating from 1 show evidence of natural selection, while slopes near 1 suggest neutrality wherein mutational selection is the driving force of codon usage patterns.

**Figure 10 viruses-13-01460-f010:**
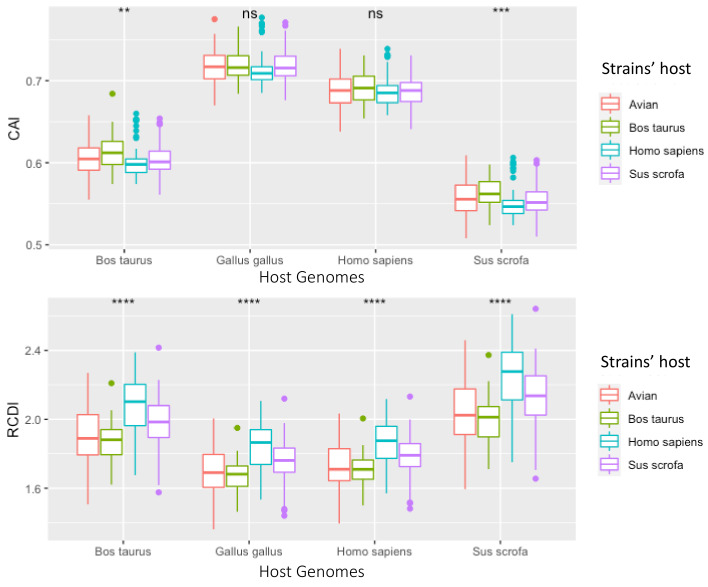
Box plots of CAI and RCDI values for each ORF from strains isolated from specific hosts. RCDI and CAI values were derived by sampling 7 complete genomes (77 segment genomes) for each host, isolated from either avian (red), cow (green), human (blue), or pig (purple) hosts. These RVA genomes were then separately compared to cow, chicken, human or pig codon usage charts. ** if *p* < 0.01, *** if *p* < 0.001, **** if *p* < 0.0001, and ns if *p* > 0.05.

**Table 1 viruses-13-01460-t001:** Association Index and Parsimony score statistics calculated in BaTS. The statistics for segments 1–11 was calculated based on 300 randomly sampled-post burn-in trees (Figure 3, Figure 4 and Figure 5) in which states for each isolate were marked as either human or non-human host. The * next to segments 4 and 9 rows at the bottom of the table is indicating the statistics are for the separately made trees for segments 4 and 9, using all available complete sequences of both VP4 and VP7 available for non-human host isolates, with the AI and PS statistics being calculated using the specific host as the state for each isolate.

Segment	Association Index (AI)	*p*-Value	Parsimony Score (PS)	*p*-Value
Segment 1	1.79	0.00	16.78	0.00
Segment 2	1.97	0.00	18.63	0.00
Segment 3	2.88	0.00	22.86	0.00
Segment 4	1.23	0.00	11.71	0.00
Segment 5	2.11	0.00	17.87	0.00
Segment 6	2.68	0.00	19.19	0.00
Segment 7	1.60	0.00	18.03	0.00
Segment 8	1.89	0.00	19.66	0.00
Segment 9	3.83	0.00	25.79	0.00
Segment 10	2.44	0.00	17.26	0.00
Segment 11	2.11	0.00	18.99	0.00
Segment 4 *	3.11	0.00	31.88	0.00
Segment 9 *	4.18	0.00	34.15	0.00

**Table 2 viruses-13-01460-t002:** Mean rates, 95% highest posterior density (HPD), coefficient of variation, and date for the ‘time to most recent common ancestor’ (TMRCA) for each segment’s phylogenetic tree.

Segment	Mean Rate	95% HPD (Lower, Upper)		Coefficient of Variation	TMRCA
Segment 1 (VP1)	1.724 × 10^−3^	1.610 × 10^−3^	1.835 × 10^−3^	2.410	1957
Segment 2 (VP2)	1.703 × 10^−3^	1.589 × 10^−3^	1.826 × 10^−3^	2.602	1963
Segment 3 (VP3)	1.948 × 10^−3^	1.814 × 10^−3^	2.087 × 10^−3^	2.436	1962
Segment 4 (VP4)	3.775 × 10^−3^	3.517 × 10^−3^	4.044 × 10^−3^	8.978	1964.5
Segment 5 (NSP1)	2.953 × 10^−3^	2.694 × 10^−3^	3.223 × 10^−3^	3.994	1964.5
Segment 6 (VP6)	1.694 × 10^−3^	1.541 × 10^−3^	1.857 × 10^−3^	3.412	1967.5
Segment 7 (NSP3)	1.615 × 10^−3^	1.454 × 10^−3^	1.778 × 10^−3^	5.061	1967
Segment 8 (NSP2)	1.473 × 10^−3^	1.335 × 10^−3^	1.610 × 10^−3^	3.323	1966.5
Segment 9 (VP7)	2.660 × 10^−3^	2.400 × 10^−3^	2.940 × 10^−3^	3.375	1966.5
Segment 10 (NSP4)	1.583 × 10^−3^	1.420 × 10^−3^	1.743 × 10^−3^	2.599	1967.5
Segment 11 (NSP5/6)	2.424 × 10^−3^	2.040 × 10^−3^	2.831 × 10^−3^	3.285	1969

## Data Availability

All genomic data analyzed in this manuscript are publicly available through NCBI’s Virus Variation Resource (https://www.ncbi.nlm.nih.gov/genome/viruses/variation/ (accessed on 20 June 2019)).

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
