# Peer review of "Rotavirus A Genome Segments Show Distinct Segregation and Codon Usage Patterns"

_viruses, 2021, doi:10.3390/v13081460_

Round 1
Reviewer 1 Report
The authors of “Rotavirus A genome segments show distinct segregation and codon usage patterns” have presented detailed analyses using more than 700 full genome sequences of Rotavirus A from the public database. They produce time-calibrated phylogenies to show how certain segments have co-segregated (e.g. segment 10 and segment 11) whilst others have more distinct evolutionary histories (e.g. segment 4 and 9) suggesting that these segments are more likely to reassort. They show by using various sequence metrics such as the Codon Adaptation Index (CAI) and the relative codon deoptimization index (RCDI) that some segments like segment 11 are likely to be better adapted to their host genomes.
The study is of interest, however the manuscript is wordy and the introduction and discussion are very repetitive. The authors also need to clarify a number of points in the methodology, provide the relevant datasets to make their results reproducible, conduct some further analysis to support some of their statements and discuss the caveats associated with sampling biases of hosts and over time.
- On line 145, you state that you used MUSCLE for doing the alignment and subsequently you have used a model in BEAST that is partitioned by codon position (line 152). As you do not mention whether you uniquely look at coding sequences or that you generated a codon alignment, it is probably inappropriate to use a model partitioned by codon position. This needs to be clarified in particular with respect to the alignment approach for segments like segment 11.
- I would recommend to share the supplementary files as tree files as the labels and dates in the word document are not legible.
- The authors should try to quantify the phylogenetic association with the host as the statement on line 232 “stricter host boundaries” is rather arbitrary. Metrics such as the Association Index (AI), Parsimony Score (PS) or maximum monophyletic clade index (MC) can be used for this purpose.
- The authors should also provide a caveat as the sampling for non-human species is relatively low which would make it less likely to detect the virus in these species and therefore harder to determining whether there is any species level pattern.
- For reproducibility, provide a table with the accession numbers used and associated sample dates or alternatively the xml files used for producing the phylogenies.
- In the discussion, the authors should discuss the difficulty of estimating evolutionary rates at deeper nodes of the phylogeny especially when the sample dates tend to be contemporaneous (there is a lot of literature on this, for example: https://doi.org/10.1093/molbev/msr266https://doi.org/10.1098/rspb.2014.0732).
- In your results, provide the range of dates used for dating.
- Figure 9. The authors have not explained how they have selected the 7 complete genomes for each host. Have they been randomly selected, has a permutation test produced?
The article is poorly punctuated and needs some thorough proof-reading to avoid sentences without verbs for example. Here is a non-extensive list of edits:
- Line 18: “conservation by host species” => unclear what is meant here
- Line 99: types also => types are also
- Sentence line 228 to 231 does not have a verb.
- The plots in figure 5 would be much easier to interpret if they were plotted as dates rather than years since 2017.
- Line 330. SI figure 12 was not available in the supplementary figures.
- Line 334: to be least => to be the least
Author Response
Dear Editor
We thank the reviewers for their interest in our work and their detailed and useful comments. We did our best to address each concern, and include their suggestions in our revised manuscript. We include a point-by-point response in red font to the reviewer comments below.
Reviewer 1 Comments:
The study is of interest, however the manuscript is wordy and the introduction and discussion are very repetitive. The authors also need to clarify a number of points in the methodology, provide the relevant datasets to make their results reproducible, conduct some further analysis to support some of their statements and discuss the caveats associated with sampling biases of hosts and over time.
- On line 145, you state that you used MUSCLE for doing the alignment and subsequently you have used a model in BEAST that is partitioned by codon position (line 152). As you do not mention whether you uniquely look at coding sequences or that you generated a codon alignment, it is probably inappropriate to use a model partitioned by codon position. This needs to be clarified in particular with respect to the alignment approach for segments like segment 11.
We clarified the partitioning scheme we used in the methods (line 156). The alignments were partitioned into coding and noncoding and the coding region was partitioned by codon position, while segment 11 did require an additional partition due to insertion events. While we did not have enough time during revisions to rerun every segment, we reran NSP5 under the partitioning scheme used previously, and we ran VP7 using no codon partitioning to see if the tree space significantly differed. We found no difference in the tree. We also assume that while the mean height/MCC annotated tree may have inaccuracies, the 350 randomly sampled trees clustering relatively closely by segment indicates a convergence to a general consensus treespace, whereas a poorly-fitting model would likely result in more spread of the segments’ trees during MDS.
2. I would recommend to share the supplementary files as tree files as the labels and dates in the word document are not legible.
The SI trees are now available as tree files along with the xml files.
3. The authors should try to quantify the phylogenetic association with the host as the statement on line 232 “stricter host boundaries” is rather arbitrary. Metrics such as the Association Index (AI), Parsimony Score (PS) or maximum monophyletic clade index (MC) can be used for this purpose.
We calculated the AI and PS using BaTS and included the results in (line 181, 249, table 1). To account for the relatively low number of non-human RVA genomes used in our main analysis, we made smaller trees for a set of isolates for just VP4 and VP7 (of all non-human isolates which included sequences for both) and calculated the AI/PS or those trees to test our presumption re VP7 having “less strict/VP4 having stricter” host boundaries. The analysis supported that assumption.
4. The authors should also provide a caveat as the sampling for non-human species is relatively low which would make it less likely to detect the virus in these species and therefore harder to determining whether there is any species level pattern.
We provided a separate tree (with the tree file and XML file available in the SI, run under a birth-death prior (rather than using tip-dates) of VP7, which used more animal strains representing every available nonhuman host-G type-date-country combination to support our suggestion of species-level patterns (figure 1, SI file 1). We chose VP7 to use for this additional tree displaying host range, as our main phylogenetic analysis was limited to complete genomes, but VP7 had more available non-human host sequences available and is thought to be more flexible to reassortment.
5. For reproducibility, provide a table with the accession numbers used and associated sample dates or alternatively the xml files used for producing the phylogenies.
We provided XML files which contain the unique strain identifier from NCBI in the tip (in this instance accession numbers are not useful as every isolate’s tip name needs to match to the other segment trees).
6. In the discussion, the authors should discuss the difficulty of estimating evolutionary rates at deeper nodes of the phylogeny especially when the sample dates tend to be contemporaneous (there is a lot of literature on this, for example: https://doi.org/10.1093/molbev/msr266 https://doi.org/10.1098/rspb.2014.0732 ).
We included a paragraph about the difficulties with estimating divergence times for viruses (line 146) due to contemporaneous bias and heterotachy: We provided several caveats to stress the inherent difficulties with estimating divergence times for viruses, however, as we were more concerned with patterns across segments rather than exact divergence dating, the sampling bias would at least be the same across all trees so we expect the general patterns would remain the same even with less sampling bias. Tip-dating usually presents more of an issue when dealing with more distantly related viruses which differ substantially in epidemiology/diverged a long time ago (so purifying selection biases the rate estimate). As rotavirus A is endemic in the population, and has been for quite some time, this somewhat minimizes the error associated with tip-dating. We excluded Avian RVA from the phylogenetic analyses because of the relatively deep branching event between Avian and mammalian RVA.
7. In your results, provide the range of dates used for dating.
We provided the range of dates (line 137 and XML/tree files)
8. Figure 9. The authors have not explained how they have selected the 7 complete genomes for each host. Have they been randomly selected, has a permutation test produced?
We included a more thorough explanation of how we chose the isolates in the method in line 226, backed up by Fig 1 and SI figs 1 and 2.
9. The article is poorly punctuated and needs some thorough proof-reading to avoid sentences without verbs for example. Here is a non-extensive list of edits:
- Line 18: “conservation by host species” => unclear what is meant here
- Line 99: types also => types are also
- Sentence line 228 to 231 does not have a verb.
- The plots in figure 5 would be much easier to interpret if they were plotted as dates rather than years since 2017.
- Line 330. SI figure 12 was not available in the supplementary figures.
- Line 334: to be least => to be the least
We corrected these and other typos and punctuation errors.
Reviewer 2 Report
This study by Hoxie and Dennehy focuses on the different evolution histories of rotavirus A segments. The analyses were done appropriately and the results are presented logically, carefully, and clearly. I have only several suggestions as follows: Major points: 1) L233: the authors explain the strict host boundaries of segment 4 with less opportunity for divergence, but I wonder if this situation can also be explained by the possibility that segment 4 is the major host determinant and the other segments are more tolerant to be reassorted. 2) The paper will be more informative if it is associated with a list of analyzed isolates, their sequence accession IDs, and the isolated years. Minor points: 3) L22: It was not clear to me what "common host genomes" indicates here. Could the authors rephrase? 4) L105: I don't know if the word "complex" can be used as a verb. 5) I found missing periods (L 439, L498) and an additional space (L527).
Author Response
Reviewer 2 Comments:
This study by Hoxie and Dennehy focuses on the different evolution histories of rotavirus A segments. The analyses were done appropriately and the results are presented logically, carefully, and clearly. I have only several suggestions as follows:
Major points:
1) L233: the authors explain the strict host boundaries of segment 4 with less opportunity for divergence, but I wonder if this situation can also be explained by the possibility that segment 4 is the major host determinant and the other segments are more tolerant to be reassorted.
We clarified this point, and included an additional AI analysis of host association by segment, suggesting segment 4 has a stronger phylogenetic association to the host (line 181, 248,, table 1).
2) The paper will be more informative if it is associated with a list of analyzed isolates, their sequence accession IDs, and the isolated years.
The trees and xml files are now available in the SI so all strain names, sequences, and dates can be easily accessed if desired.
Minor points:
3) L22: It was not clear to me what "common host genomes" indicates here. Could the authors rephrase?
We rephrased to improve the clarity of this point (L22).
4) L105: I don't know if the word "complex" can be used as a verb.
We changed “complex” to “associates.”
5) I found missing periods (L 439, L498) and an additional space (L527).
We fixed these typos.